# Association of urban inequality and income segregation with COVID-19 mortality in Brazil

J. Firmino de Sousa Filho[1,2]☯*, Uriel M. Silva[3]☯, Larissa L. Lima[3,4]☯, Aureliano S. S. Paiva[1]☯, Gervásio F. Santos[1,2]☯, Roberto F. S. Andrade[1,5]☯, Nelson Gouveia[6]☯, Ismael H. Silveira[1,7]☯, Amélia A. de Lima Friche[3]☯, Maurício L. Barreto[1,7]☯, Waleska Teixeira Caiaffa[3]☯

1 Center of Data and Knowledge Integration for Health (CIDACS), Salvador, Brazil, 2 Faculty of Economics (PPGE)–Federal University of Bahia, Salvador, Brazil, 3 Observatory for Urban Health in Belo Horizonte (OSUBH)–Federal University of Minas Gerais, Belo Horizonte, MG, Brazil, 4 Postgraduate Program in Mathematical and Computational Modeling–Federal Center for Technological Education of Minas Gerais (CEFET-MG), Belo Horizonte, MG, Brazil, 5 Institute of Physics–Federal University of Bahia, Salvador, Brazil, 6 University of São Paulo Medical School, São Paulo, Brazil, 7 Institute of Public Health (ISC)–Federal University of Bahia, Salvador, Brazil

☯ These authors contributed equally to this work.
* jose.sousa@ufba.br

**Data Availability Statement:** The SALURBAL project welcomes queries from anyone interested in learning more about its dataset and potential access to data. To learn more about SALURBAL's

## Abstract

Socioeconomic factors have exacerbated the impact of COVID–19 worldwide. Brazil, already marked by significant economic inequalities, is one of the most affected countries, with one of the highest mortality rates. Understanding how inequality and income segregation contribute to excess mortality by COVID–19 in Brazilian cities is essential for designing public health policies to mitigate the impact of the disease. This paper aims to fill in this gap by analyzing the effect of income inequality and income segregation on COVID–19 mortality in large urban centers in Brazil. We compiled weekly COVID–19 mortality rates from March 2020 to February 2021 in a longitudinal ecological design, aggregating data at the city level for 152 Brazilian cities. Mortality rates from COVID-19 were compared across weeks, cities and states using mixed linear models. We estimated the associations between COVID-19 mortality rates with income inequality and income segregation using mixed negative binomial models including city and week-level random intercepts. We measured income inequality using the Gini index and income segregation using the dissimilarity index using data from the 2010 Brazilian demographic census. We found that 88.2% of COVID–19 mortality rates variability was between weeks, 8.5% between cities, and 3.3% between states. Higher-income inequality and higher-income segregation values were associated with higher COVID–19 mortality rates before and after accounting for all adjustment factors. In our main adjusted model, rate ratios (RR) per 1 SD increases in income inequality and income segregation were associated with 17% (95% CI 9% to 26%) and 11% (95% CI 4% to 19%) higher mortality. Income inequality and income segregation are long-standing hallmarks of large Brazilian cities. Risk factors related to the socioeconomic context affected the course of the pandemic in the country and contributed to high mortality rates. Pre-existing social vulnerabilities were critical factors in the aggravation of COVID–19, as supported by the observed associations in this study.

dataset, visit https://drexel.edu/lac/ or contact the project at salurbal@drexel.edu.

**Funding:** The Salud Urbana en América Latina (SALURBAL)/Urban Health in Latin America project is funded by the Wellcome Trust [205177/Z/16/Z]. JFSF, UMS, LLL, ASSP, GFS, AALF, NG, WTC are supported by Wellcome Trust [205177/Z/16/Z]. The funders had no role in study design, data collection and analysis, decision to publish, or preparation of the manuscript. <https://wellcome.org/>.

**Competing interests:** The authors have declared that no competing interests exist.

## Introduction

On January 30, 2020, the World Health Organization (WHO) declared that the new Coronavirus (SARS-CoV-2) outbreak constituted a global public health emergency. In this context, WHO sought to improve coordination and cooperation mechanisms at an international level to contain the spread of the virus. On March 11, 2020, COVID–19 was characterized as a pandemic disease, which means that it was already present on all continents [1, 2]. The way each country dealt with the pandemic reflected their particular political, economic, and social context. Populations in more vulnerable socioeconomic conditions were disproportionately exposed to infection since they had more barriers to maintaining social isolation and frequently had to work in relatively unsafe conditions [3–6].

The Americas have become a cause for concern due to the inequality and poverty prevalent in the region, especially in the low and middle-income countries in Latin America. Understanding the relationships between inequalities and urban health is a complex challenge for which no single scientific discipline can offer a complete solution. Income concentration and segregation directly impact the health and the well-being of populations living in cities [7–10]. The accelerated and usually unplanned population growth in large cities exacerbates these relationships. Historically, societies change through social and economic crises and epidemiological and sanitary ones [11].

Brazil is the largest and one of the most unequal countries in Latin America. Inequality in Brazil has persisted over time [12]. Further, there is a close relationship between inequality and income segregation in the country because, typically, the magnitude of differences in access to resources and status across the population directly correlates with the degree of spatial segregation [13, 14]. Despite this relationship, there are important differences between inequality and income segregation. Inequality reflects issues related to the overall allocation of resources in a society, while segregation deals with the spatial dimension of this allocation problem [7, 15]. Segregation is materialized in the urban environment and becomes a geographic expression of inequality between different social groups in Brazilian cities. Notably, this results in the isolation of economically vulnerable people and weakens access to urban public services as well as health [16].

Brazil is one of the countries most affected by the pandemic caused by SARS-CoV-2 [17]. Several factors contributed to the rapid and critical spread of COVID–19 in the country and, consequently, the excessive number of cases and deaths. Among these factors were structural social and environmental causes such as income inequality, poverty, segregation, and lack of access to adequate healthcare [18, 19]. These elements have intensified the effects of the pandemic, observed not only in Brazil but also throughout Latin America [3, 20]. In the context of the current pandemic, much attention has been given to vulnerabilities related to health systems, whether for prevention or care of COVID–19 morbidity and mortality. In Brazil, a universal and free healthcare system (the Unified Health System, or SUS in the original Portuguese acronym) was essential to provide health care for the disease. However, in some country areas, the SUS showed signs of collapse [10, 16, 20].

This study investigates the association of income inequality and income segregation with COVID–19 mortality rates in large Brazilian urban agglomerations. The main research question is framed as: "how are urban income inequalities and segregation among Brazilian cities associated with COVID–19 mortality rates?" We hypothesize that both inequality and segregation increase COVID–19 occurrence and mortality rates. Spatial segregation of low-income families in cities may hinder the effectiveness of policies aiming to reduce COVID–19 incidence. Therefore, we provide evidence that might identify settings with the need for further containment strategies and interventions to mitigate the impacts of future pandemics.

## Materials and methods

This study is part of the SALURBAL (*Salud Urbana en América Latina*, in the original Spanish acronym) Project, an international collaboration investigating the health impacts of urban policies and urban environment features in cities across 11 countries in Latin America and the Caribbean [21]. "Cities" within the Project are defined as aggregates of country-specific administrative units, referred to as subcities (*municipios* in Brazil). The selection criterion for cities is that they should have at least 100,000 inhabitants within a contiguous built-up urban area, identified using satellite imagery [22]. The sample for this study is then comprised of 422 subcities nested within 152 Brazilian cities, since this is the total number of contiguous built-up urbean areas with more than 100,000 inhabitants identified by the project. This set of cities had a total of 121 million inhabitants, approximately 63% of the country's population in 2010.

### Data sources

Our outcome variable is weekly COVID–19 mortality rates per 100,000 inhabitants in cities. Data on COVID–19 death counts was obtained from Cota (2020) [23], which compiles information from both the official influenza surveillance and the mortality information systems, i.e. the *Sistema de Informação da Vigilância Epidemiológica da Gripe*, (SIVEP-Gripe) and *Sistema de Informação de Mortalidade* (SIM) *(*both in the original Portuguese acronym). Data include individuals who died inside or outside hospital care. Deaths were recorded daily from March 23, 2020, to February 26, 2021 (only aggregate records for each city were available; the dataset does not have individual data). For the denominators, 2019 population estimates were obtained by projecting the population from the 2010 demographic census for each city [24].

The exposures in this study were the income-based Gini index for income inequality and a dissimilarity index for income segregation, both defined at the city level. The Gini and the dissimilarity index were computed for each city using income data from the 2010 census conducted by the Brazilian Statistical Bureau (IBGE, Portuguese acronym) [25, 26]. Both indexes range from 0 to 1. When the Gini index is close to 0, it indicates greater income equality between families. In contrast, the closer it is to 1, the greater the income inequality. The dissimilarity index considered the cut-off of families earning up to 2 minimum wages, as previously established in other studies for SALURBAL cities in Brazil [14]. Income segregation is at its minimum value (0) when income groups are evenly distributed across neighborhoods, or all spatial units have an equal proportion of the different income groups. Segregation achieves its maximum value of 1 when the census tracts or neighborhoods of the city have no internal variation, with all of each spatial unit's residents belonging to the same income group.

Other variables included the percentage of the population aged > 65 years old, hospital beds, general practitioners and nurses per capita; percentage of the population with private health insurance and the population below the poverty line; GDP per capita; and educational attainment score were included as adjustment factors in the association between COVID-19 mortality and income inequality and segregation. All of these variables were defined at the city level, and were time-invariant.

GDP was obtained from the 2010 IBGE census estimates [27] and converted to 2010 purchasing power parity US dollars. The education score, also extracted from the 2010 Brazilian Census, was defined as the sum of the Z-score of the percentage of the population older than 25 years of age with at least secondary education and the Z-score of the percentage of the population older than 25 years of age with university education. Higher score values indicate better educational achievement in the population [28].

The poverty index was computed as the proportion of households below the poverty line (defined as having a per capita income of less than half of the minimum wage), also based on 2010

census data [29]. The population aged 65 years or above was derived from the 2019 intercensal population projections. The proportion of people with private health insurance was obtained from the national insurance bureau (ANS, *Agência Nacional de Saúde Suplementar* in the original Portuguese acronym). The other healthcare indicators came from the Brazilian Ministry of Health. For further details on the definition and sources of all the variables used in this study, see S1 Table. All variables used in this study were compiled and harmonized by the SALURBAL team [21, 22].

## Statistical analysis

First, we computed weekly mortality rates due to COVID–19 per 100,000 inhabitants for each city by dividing the corresponding number of deaths by the 2019 population projections and multiplying by 100,000. "Week" is the period defined for each city as the seven days following the second recorded COVID–19 case in that city. Therefore, week 1 is equal to the 1–7 days after the second recorded case in the city; week 2 is the 8–15 days after the second recorded case, and so on for the entire sample period of March 23, 2020 to February 26, 2021.

In order to increase comparability across cities, we have "balanced" our sample by removing, for each city, any weeks exceeding the number of weeks available for the city with the fewest number of weeks in the sample. The cities of *Barreiras* and *Parobé*, in particular, had only 41 weeks of data available in the sudied period, the smallest number across all 152 cities. This means for example that a city such as São Paulo, which had a total of 49 weeks of data available (the maximum possible for the period), had its additional 8 weeks removed so that only its first 41 weeks of data were taken into account for this study.

Second, we described the distribution of all variables included in the study across cities. We also decomposed the variability in COVID–19 mortality rates by fitting a multilevel linear model with the logarithm of weekly COVID–19 mortality rates as the outcome with random intercepts for each week, city and "state" (states are Brazilian federative units).

Finally, we investigated the associations between COVID–19 mortality rates and the exposures in a series of longitudinal multilevel negative binomial models with random intercepts for each city and week. The associations with each exposure were estimated separately because of the high correlation among them (0.8) (see the correlation matrix in S1 Fig). We developed a causal directed acyclic graph (DAG) (S2 Fig) to illustrate the hypotheses under study and to select a minimal sufficient set of adjustment variables that would allow the identification of the direct effect of inequality and income segregation on COVID–19 mortality. Then, we estimate three models: the first one is the unadjusted model, the second one estimates the total effect (our main model), and the third model measures the direct effect (not mediated by access to health care). Using the DAG we identified the following variables as necessary for adjustment: percentage of the population aged 65 years old or higher, healthcare indicators, poverty, GDP per capita, and educational attainment score. Ultimately, poverty had to be excluded from the set of adjustment covariates due to multicollinearity effects. (See S4 and S5 Figs for the other variables). Since our source of COVID-19 mortality only included aggregate deaths, we cannot adjust for individual sex and age.

For all negative binomial models, weekly COVID-19 death counts per 100,000 inhabitants was the outcome variable, and the offset was the logarithm of the 2019 population projections (expressed in 100,000 inhabitants). Before fitting the models, all exposures and covariates were transformed into Z-scores. All analyses were conducted with the package glmmTMB [30] in the R software environment.

## Results

Table 1 contains summary statistics of the variables used in the study. Across all cities, average weekly COVID–19 mortality rates was 2.47 (SD 0.91) per 100,000 inhabitants. Mean income

**Table 1. Summary statistics of all variables used in this study.**

| Variables | Mean | Standard deviation |
|---|---|---|
| **Weekly COVID–19 mortality rates** | 2.47 | (0.91) |
| **Income inequality (Gini index)** | 0.55 | (0.05) |
| **Income segregation (dissimilarity index)** | 0.27 | (0.05) |
| **Population aged $\geq$ 65 years (%)** | 9.44 | (2.26) |
| **GDP per capita (2010 PPP USD)** | 16,662.89 | (9,915.14) |
| **Education attainment score** | -0.19 | (1.17) |
| **Poverty index (%)** | 25.44 | (13.28) |
| **Hospital beds per capita[a]** | 241.93 | (102.23) |
| **General practitioners per capita[a]** | 242.3 | (107.19) |
| **Nurses per capita[a]** | 144.38 | (51.9) |
| **Population with private health insurance (%)** | 26.08 | (11.04) |
| **Total city population** | 799,902.49 | (2,129,242.78) |

[a]Per capita indicators were multiplied by 100,000 prior to computing the summary statistics in this table.
Income inequality, income-based segregation, GDP per capita, poverty index and the educational attainment score are computed using data from the IBGE 2010 census. Data on healthcare (number of hospital beds, number of general practitioners, population with private health insurance) and demographics (population aged $\geq$ 65 years old) are from 2019.

inequality was 0.55 (SD 0.05) and mean income segregation was 0.27 (SD 0.05). The population aged 65 years or older occupy in average a proportion of 9.44% (SD 2.26%) of the total population. Average GDP per capita was USD 16,662.89 (SD 9,915.14). The educational attainment score's mean across cities was -0.19 (SD 1.17), and the poverty index's mean was 25.44% (SD 13.28%). The average of the population in poverty situation was 24.4%.

Table 2 contains the variance estimates for the random intercepts of each level in the linear mixed model with weekly log-COVID–19 mortality rates as the outcome. Most of the variability (88.19%) in the log-mortality rates was between weeks. About 8.52% of the variability was between cities, and 3.29% between states.

The spatio-temporal distribution of COVID–19 mortality rates across cities, aggregated over a period of weeks, is shown in S3 Fig From March 2020 to June 2020, the risk of death by COVID–19 was up to 10 times larger in the state capitals, especially those close to the Brazilian coast. Later in 2020, mortality rates increased throughout the entire country, including in the central and non-coastal areas.

Table 3 contains the estimated associations between weekly COVID–19 mortality rates and both exposures considered in this study. In the second model, a 1SD increase in income

**Table 2. Variance estimates and percentage of total variance of COVID–19 weekly mortality rates per 100,000 inhabitants for week, city, state and region level in a linear mixed model.**

| Level | Variance Estimate | Percentage of Total Variance |
|---|---|---|
| **Week** | 0.4007 | 88.19% |
| **City** | 0.0387 | 8.52% |
| **State** | 0.0150 | 3.29% |

Estimates correspond to the variances of the random intercepts for each level. The model has the logarithm of COVID–19 weekly mortality rates per 100,000 inhabitants as outcome, and includes random intercepts for each week, city and state. We add "1" to each rate prior to log-transforming to ensure that the log-rates are well-defined when the rates are very small or equal to zero.

**Table 3. Estimated associations (rate ratios and 95% confidence intervals) between city weekly COVID–19 mortality rates and the exposures and adjustment factors using mixed negative binomial regression models.**

| Variables | Model 1 | Model 2 | Model 3 |
|---|---|---|---|
| **Models with income inequality as the exposure** | | | |
| **Income inequality** | 1.13 (1.07–1.20) | 1.17 (1.09–1.26) | 1.16 (1.08–1.25) |
| **Population aged $\geq$ 65 years** | .. | 1.00 (0.93–1.07) | 1.02 (0.94–1.10) |
| **GDP per capita** | .. | 1.04 (0.98–1.11) | 1.06 (0.98–1.14) |
| **Education attainment** | .. | 0.91 (0.85–0.98) | 0.95 (0.87–1.04) |
| **Hospital beds per capita** | .. | .. | 1.05 (0.96–1.14) |
| **General practitioners per capita** | .. | .. | 0.98 (0.88–1.09) |
| **Nurses per capita** | .. | .. | 0.95 (0.87–1.04) |
| **Private health insurance** | .. | .. | 0.95 (0.87–1.04) |
| **Models with income segregation as the exposure** | | | |
| **Income segregation** | 1.11 (1.04–1.18) | 1.11 (1.04–1.19) | 1.10 (1.03–1.18) |
| **Population aged $\geq$ 65 years** | .. | 0.97 (0.90–1.04) | 0.98 (0.90–1.06) |
| **GDP per capita** | .. | 1.03 (0.96–1.10) | 1.04 (0.96–1.12) |
| **Education attainment** | .. | 0.97 (0.90–1.03) | 1.00 (0.92–1.10) |
| **Hospital beds per capita** | .. | .. | 1.07 (0.98–1.17) |
| **General practitioners per capita** | .. | .. | 0.99 (0.89–1.10) |
| **Nurses per capita** | .. | .. | 0.93 (0.84–1.02) |
| **Private health insurance** | .. | .. | 0.96 (0.88–1.06) |

The outcome is weekly COVID–19 mortality rates for each city. All exposures and adjustment factors are time-invariant, and are also defined at the city level. All models assume a Negative Binomial distribution with constant dispersion parameter and include random intercepts for each city and week. A logarithmic link was used, and for properly expressing the outcome as rates, the dependent variable was COVID–19 death counts and an offset equal to the log of the corresponding 2019 population projections (per 100,000) was used. All exposures and adjustment factors were standardized to have a mean of 0 and a standard deviation of 1, so that interpretation of the associations is always with respect to a 1SD increase in the corresponding exposure/adjustment.
For each exposure, Model 1 is the unadjusted model, Model 2 includes all of the adjustment factors with the exception of the healthcare indicators (total effect), and Model 3 further includes the healthcare indicators (direct effect).

inequality was associated with a 1.17 (CI 1.09–1.26) increase in COVID–19 mortality rates, and a 1SD increase in income segregation was associated with a 1.11 (CI 1.04–1.19) increase in COVID–19 mortality rates. In the fully adjusted model (i.e. after further adjusts for possible mediating effects of health care), both income inequality (RR 1.16, CI 1.08–1.25) and segregation (RR 1.10, CI 1.03–1.18) remained positively associated with COVID–19 mortality rates.

In order to assess collinearity between exposures and covariates, we have computed Variance Inflation Factors (VIF) for both fully-adjusted models (S4 and S5 Figs). Although we initially suspected that healthcare indicators might be collinear, this is actually not true: for the model with Gini as the exposure, VIF ranged from 1.23 to 1.77, and for the model with segregation as the exposure, VIF ranged from 1.11 to 1.77.

Finally, we also performed a sensitivity analysis in order to assess whether the estimated associations in the negative binomial models are affected by whether we "balance" the sample beforehand or not. S6 Fig contains a forest plot of the estimated associations between COVID-19 mortality rates and inequality or segregation across all estimated models, without balancing the sample. Although point estimates and confidence intervals are numerically different, there are no qualitative changes to the main results presented in Table 3.

## Discussion

Higher income inequality and segregation are associated with COVID-19 mortality rates in large Brazilian cities. As inequality and income segregation increase by 1SD, the risk of dying from COVID-19 increased 17% (CI 1.09–1.26) and 11% (CI 1.04–1.19), respectively. Therefore, our findings suggest that the more unequal and segregated cities suffered a more significant impact from the COVID–19 pandemic. Populations often subject to inadequate living conditions are more vulnerable to covid-19 morbidity and mortality [31]. Reasons for this may include barriers to social distancing in daily activities (e.g., commuting to work), since more unequal and segregated areas also tend to have high population density or household overcrowding, especially in informal settlements and slums [32].

The relationship between COVID-19 mortality rates and the covariates used in our study, on the other hand, was not significant for any of the estimated models. However, since the association between COVID-19 with inequality and segregation is influenced by different social aspects and families' living conditions and well-being (here captured by overall income, education, and access to essential health services), these are still theoretically relevant features to control the associations.

Following the protocols of lockdown and social distancing was difficult for the population in poorer countries. One cause for this is that the economic and social support for this population does not exist or lasted only for a short time, forcing individuals to go out searching for jobs, take overcrowded public transport, and be exposed to various unhealthy conditions [33]. Income inequality and segregation limit the options for overcoming common barriers due to lack of resources or due to the spatial isolation of the most vulnerable populations. Thus, following public health recommendations was not a reality since a large part of the population (e.g. informal workers or those looking for a job) could not stay at home [34]. In Brazil, this effect is even worse because there is a widespread denial of the pandemic, reflecting the government's lack of engagement with the problem and its lack of interest in proposing actions based on scientific evidence to reduce the spread of the disease [31].

High-income inequality undermines social cohesion and erodes the population's confidence in its government, affecting the overall responsiveness to sanitary crises [32]. Historically, fragmented political actions and overall lack of coordination by the Federal Government with other institutions was a key factor in promoting inequalities in Brazil. Additionally, the same factors, led the country to attain one of the highest COVID-19 incidence and mortality rates in the world. The greater the income inequality and segregation is, the worse the conditions for access to healthcare, which significantly increases vulnerability to COVID–19. Crucially, inequality affects the vast majority of the population, not just the poor [35]. Therefore, since in more unequal or more segregated cities the inhabitants are typically at a higher risk for COVID-19 deaths, it is essential to strengthen control measures and implement coordinated actions to mitigate the pandemic in these areas.

### Strengths and limitations

The longitudinal design and the multilevel modeling approach used to estimate the associations that were investigated in this study have the advantage of accommodating unobserved heterogeneity in COVID–19 mortality rates between cities, which arises due to a variety of reasons, e.g. underreporting, reporting delays and misdiagnosis. Further, our definition of "weeks" was selected to enhance comparability across the different stages of the pandemic.

As for the limitations, the ecologic structure of the data does not allow for inference at the individual level, and the results obtained for the large areas considered for the study (all with ≥ 100,000 residents) might not generalize to smaller Brazilian municipalities. We also

have no time-varying exposures, making it impossible to draw any conclusions regarding the temporal dynamic effects of inequality and segregation in COVID–19 mortality throughout 2020. The lack of individual-level death records also meant that we cannot account for individual age and sex-specific effects on mortality rates, having to rely on contextual effects instead (e.g., the proportion of the population aged 65 years old or higher in each city).

Another limitation is that income data by census tract level used to compute the dissimilarity index were collected by the Brazilian Bureau of statistics in 2010 (the last nationwide census; the 2020 demographic census has been postponed to 2022 because of the COVID-19 pandemic). Thus, these socioeconomic data may not reflect the current reality of the population.

Further, we chose to focus on the associations between structural inequality and segregation and COVID–19 prior to the introduction of vaccination policies. Future work should examine whether the associations observed here are consistent in more recent waves of the pandemic and consider vaccination data.

The negative binomial model used to estimate the associations between COVID-19 mortality rates and income inequality and segregation also can result in some limitation. In these models, a basic underlying assumption is that the relationship between the outcome and the exposures are linear (log-linear to be more specific, since a logarithmic link is being used). We did a brief exploration of this assumption through the use of scatterplots and LOESS fits (see S7 and S8 Figs), and saw no departures from linearity in the relationship between log-COVID-19 mortality rates and either income inequality or segregation.

## Conclusions

The first year of the COVID–19 pandemic in Brazil provided us with essential insights into the serious structural problems in the country's large urban centers. Our objective in this study was to highlight the wide socioeconomic inequalities (a hallmark of Brazilian cities) and emphasize a pre-existing vulnerable structure that should be a priority for future public policies aiming at mitigating the damage done by the spread of COVID–19 infection in the country. Despite its limitations, this study provides consistent evidence that inequality and income segregation are adversely associated with city-wide mortality due to COVID–19. The crucial lessons learned from how Brazil's fragile socioeconomic conditions and inequalities are related to COVID–19 mortality should be shared with and compared with other countries in the world, especially those in Latin America and those with similar economic institutions. This experience will be crucial for building cooperation policies and strategies to help prevent new health tragedies.

## Supporting information

**S1 Table. Overall structure of the data used in this paper.** Each variable was collected and harmonized by SALURBAL's team based on data from the corresponding source. *SIVEP-Gripe: *Sistema de Informação da Vigilância Epidemiológica da Gripe* (original Portuguese acronym). **SIM: *Sistema de Informação de Mortalidade* (original Portuguese acronym). ***ANS: *Agência Nacional de Saúde Suplementar* (original Portuguese acronym), at www.ans.gov.br. ****The publicly available data platform of the Brazilian Ministry of Health is www.tabnet.datasus.gov.br. Complete data and official publications regarding the 2010 IBGE (*Instituto Brasileiro de Geografia e Estatística*, in the original Portuguese acronym) [9]. Census can be found at https://censo2010.ibge.gov.br/.
(XLSX)

**S1 Fig. Correlation matrix between all of the exposures and covariates used in the paper.**
(PDF)

**S2 Fig. Direct acyclic graph (DAG)\* of the causal associations and relationships between income inequality and income segregation with COVID-19 mortality.** \*DAG built using DAGitty®.
(PDF)

**S3 Fig. City-level COVID–19 mortality rates, aggregated across weeks.**
(PDF)

**S4 Fig. Variance inflation factors of the exposure and covariates for the fully adjusted negative binomial model estimating the association between Gini and COVID-19 mortality.**
(PDF)

**S5 Fig. Variance inflation factors of the exposure and covariates for the fully adjusted negative binomial model estimating the association between segregation and COVID-19 mortality.**
(PDF)

**S6 Fig. Forest plot of estimated associations between COVID-19 mortality rates and income inequality and segregation for no-balancing sample.** Dots are point estimates (rate ratios), and confidence bars are 95% confidence intervals. Model 1 = unadjusted model, Model 2 = adjusted by social environment covariates, Model 3 = fully adjusted model (social environment covariates and healthcare indicators).
(PDF)

**S7 Fig. Scatterplots of the logarithm of COVID-19 mortality rates (per 100,000 hab., aggregated across all weeks) versus the Gini index for each city.** The solid black line represents the LOESS fit, with its 95% confidence interval given in the shaded bands. CO = *Centro-Oeste* (Central-West), N = *Norte* (North), NE = *Nordeste* (Northeast), S = *Sul* (South), SE = *Sudeste* (Southeast).
(PDF)

**S8 Fig. Scatterplots of the logarithm of COVID-19 mortality rates (per 100,000 hab., aggregated across all weeks) versus segregation for each city.** The solid black line represents the LOESS fit, with its 95% confidence interval given in the shaded bands. CO = *Centro-Oeste* (Central-West), N = *Norte* (North), NE = *Nordeste* (Northeast), S = *Sul* (South), SE = *Sudeste* (Southeast).
(PDF)

**S1 File. "Association of urban inequality and income segregation with COVID-19 mortality in Brazil".**
(DOCX)

## Author Contributions

**Conceptualization:** J. Firmino de Sousa Filho, Uriel M. Silva, Gervásio F. Santos, Roberto F. S. Andrade, Nelson Gouveia, Amélia A. de Lima Friche, Maurício L. Barreto, Waleska Teixeira Caiaffa.

**Data curation:** J. Firmino de Sousa Filho, Uriel M. Silva, Amélia A. de Lima Friche, Waleska Teixeira Caiaffa.

**Formal analysis:** J. Firmino de Sousa Filho, Uriel M. Silva, Larissa L. Lima, Aureliano S. S. Paiva, Nelson Gouveia, Ismael H. Silveira, Amélia A. de Lima Friche, Maurício L. Barreto, Waleska Teixeira Caiaffa.

**Investigation:** J. Firmino de Sousa Filho, Uriel M. Silva, Larissa L. Lima, Nelson Gouveia, Maurício L. Barreto, Waleska Teixeira Caiaffa.

**Methodology:** J. Firmino de Sousa Filho, Uriel M. Silva, Larissa L. Lima, Aureliano S. S. Paiva, Roberto F. S. Andrade, Ismael H. Silveira, Waleska Teixeira Caiaffa.

**Software:** J. Firmino de Sousa Filho, Uriel M. Silva, Roberto F. S. Andrade.

**Supervision:** Gervásio F. Santos, Roberto F. S. Andrade, Nelson Gouveia, Ismael H. Silveira, Amélia A. de Lima Friche, Maurício L. Barreto, Waleska Teixeira Caiaffa.

**Validation:** Larissa L. Lima, Aureliano S. S. Paiva, Gervásio F. Santos, Roberto F. S. Andrade, Nelson Gouveia, Ismael H. Silveira, Amélia A. de Lima Friche, Maurício L. Barreto, Waleska Teixeira Caiaffa.

**Visualization:** Larissa L. Lima, Aureliano S. S. Paiva, Gervásio F. Santos, Roberto F. S. Andrade, Nelson Gouveia, Ismael H. Silveira, Maurício L. Barreto.

**Writing – original draft:** J. Firmino de Sousa Filho, Uriel M. Silva, Larissa L. Lima, Aureliano S. S. Paiva, Gervásio F. Santos, Roberto F. S. Andrade, Nelson Gouveia, Ismael H. Silveira, Amélia A. de Lima Friche, Maurício L. Barreto, Waleska Teixeira Caiaffa.

**Writing – review & editing:** J. Firmino de Sousa Filho, Uriel M. Silva, Larissa L. Lima, Aureliano S. S. Paiva, Gervásio F. Santos, Roberto F. S. Andrade, Nelson Gouveia, Ismael H. Silveira, Amélia A. de Lima Friche, Maurício L. Barreto, Waleska Teixeira Caiaffa.

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
