## [Decision Letter · Decision Letter 0]

19 Aug 2022

PONE-D-22-17397Association of urban inequality and income segregation with COVID-19 mortality in BrazilPLOS ONE

Dear Dr. Sousa Filho,

Thank you for submitting your manuscript to PLOS ONE. After careful consideration, we feel that it has merit but does not fully meet PLOS ONE’s publication criteria as it currently stands. Therefore, we invite you to submit a revised version of the manuscript that addresses the points raised during the review process. Your manuscript has been assessed by one peer-reviewer, and their comments are appended below.  The reviewer has asked for some additional clarification regarding the methodology of the study, for example regarding the variables used and the justification of the sample selection. In addition, the reviewer states that the manuscript requires a more in-depth discussion of the limitations to this study.  Could you please carefully revise the manuscript to address all comments raised? Please note that we have only been able to secure a single reviewer to assess your manuscript. We are issuing a decision on your manuscript at this point to prevent further delays in the evaluation of your manuscript. Please be aware that the editor who handles your revised manuscript might find it necessary to invite additional reviewers to assess this work once the revised manuscript is submitted. 

We look forward to receiving your revised manuscript.

Kind regards,

Maria Elisabeth Johanna Zalm, Ph.D

Editorial Office

PLOS ONE

Journal Requirements:

“The Salud Urbana en América Latina (SALURBAL)/Urban Health in Latin America project is funded by the Wellcome Trust [205177/Z/16/Z].”

“The Salud Urbana en América Latina (SALURBAL)/Urban Health in Latin America project is funded by the Wellcome Trust [205177/Z/16/Z].

JFSF, UMS, LLL, ASSP, GFS, AALF, NG, WTC are supported by Wellcome Trust [205177/Z/16/Z].

https://wellcome.org/”

Reviewers' comments:

Reviewer's Responses to Questions

**Comments to the Author**

1. Is the manuscript technically sound, and do the data support the conclusions?

Reviewer #1: Yes

2. Has the statistical analysis been performed appropriately and rigorously? 

Reviewer #1: Yes

3. Have the authors made all data underlying the findings in their manuscript fully available?

Reviewer #1: Yes

4. Is the manuscript presented in an intelligible fashion and written in standard English?

Reviewer #1: Yes

5. Review Comments to the Author

Reviewer #1: This study documented that more unequal Brazilian cities suffered higher mortality from covid-19. The theme is relevant to global public health; findings reported here can instruct health policy and planning. This reviewer agrees with the main methodological options and shares some specific concerns.

1. Using 2010 data on socioeconomic status (income, education, etc.) is the most relevant unacknowledged study limitation. I understand that Brazil may not have performed a more recent census, which limits the current study and must be discussed.

2. The study outcome variable is weekly covid-19 death rates. The pandemic took a higher toll on specific gender and age groups; furthermore, city-level socioeconomic status may be a function of its proportion of older people. This observation account for the importance of adjusting death rates for age and gender. Please inform the reader if it was done; please justify if it was not.

3. I also missed explaining or justifying the sample of 152 Brazilian cities. I understood that a previous study gathered the database. However, the reader should be able to understand the study without having to search for previous publications. Please inform the reader what criteria guided the selection of these cities.

4. Except for the primary exposures (income inequality and income segregation), the remaining covariates did not associate with the outcome in Table 3’s models. This finding was not discussed.

5. In the Abstract, the authors concluded (and emphasized) that “socioeconomic inequalities can be reduced permanently through consistent long-term policies.” As it stands, this statement is just wishful thinking. This study has not assessed the effectiveness of “consistent long-term policies” permanently reducing socioeconomic inequalities. It is curious to observe that, in the main text, the conclusion did not repeat this argument.

6. PLOS authors have the option to publish the peer review history of their article (what does this mean?). If published, this will include your full peer review and any attached files.

Reviewer #1: **Yes: **Jose Leopoldo Ferreira Antunes

---

## [Author Response · Author response to Decision Letter 0]

2 Sep 2022

Dear reviewer, we thank you for your valuable comments and hope that we have answered all of them concisely. The changes are highlighted in red in the “manuscript with tracked changes” document.

Review Comments to the Author

Reviewer #1: This study documented that more unequal Brazilian cities suffered higher mortality from covid-19. The theme is relevant to global public health; findings reported here can instruct health policy and planning. This reviewer agrees with the main methodological options and shares some specific concerns.

1. Using 2010 data on socioeconomic status (income, education, etc.) is the most relevant unacknowledged study limitation. I understand that Brazil may not have performed a more recent census, which limits the current study and must be discussed.

Thank you for your comment. We have inserted this paragraph in the “strengths and limitations” section to clarify this fragility.

“Another limitation is that income data by census tract level used to compute the dissimilarity index were collected by the Brazilian Bureau of statistics in 2010 (the last nationwide census; the 2020 demographic census has been postponed to 2022 because of the COVID-19 pandemic). Thus, these socioeconomic data may not reflect the current reality of the population.”

2. The study outcome variable is weekly covid-19 death rates. The pandemic took a higher toll on specific gender and age groups; furthermore, city-level socioeconomic status may be a function of its proportion of older people. This observation account for the importance of adjusting death rates for age and gender. Please inform the reader if it was done; please justify if it was not.

Thank you for your comment. We have inserted this sentence in the first paragraph of the “Data sources” subsection: “(only aggregate records for each city were available; the dataset does not have individual data)”. 

To further clarify the issue, we have also inserted this sentence in the “Statistical analysis” subsection: “Since our source of COVID-19 mortality only included aggregate deaths, we cannot adjust for individual sex and age”. 

Finally, we acknowledge this properly in the “Strengths and limitations” subsection, with the sentence “The lack of individual-level death records also meant that we cannot account for individual age and sex-specific effects on mortality rates, having to rely on contextual effects instead (e.g., the proportion of the population aged 65 years old or higher in each city)”.

3. I also missed explaining or justifying the sample of 152 Brazilian cities. I understood that a previous study gathered the database. However, the reader should be able to understand the study without having to search for previous publications. Please inform the reader what criteria guided the selection of these cities.

Thank you for your comment, we have complemented the following sentence at the end of the “materials and methods” section: 

 “The selection criterion for cities is that they should have at least 100,000 inhabitants within a contiguous built-up urban area, identified using satellite imagery [22].”

“In addition, 121 million people live in these urban agglomerations in Brazil or approximately 63% of the total population in 2010.”

4. Except for the primary exposures (income inequality and income segregation), the remaining covariates did not associate with the outcome in Table 3’s models. This finding was not discussed.

Thank you for this suggestion. We have inserted the following in the talk section to address this comment: 

“The relationship between COVID-19 mortality rates and the covariates used in our study, on the other hand, was not significant for any of the estimated models. However, since the association between COVID-19 with inequality and segregation is influenced by different social aspects and families' living conditions and well-being (here captured by overall income, education, and access to essential health services), these are still theoretically relevant features to control the associations.”

5. In the Abstract, the authors concluded (and emphasized) that “socioeconomic inequalities can be reduced permanently through consistent long-term policies.” As it stands, this statement is just wishful thinking. This study has not assessed the effectiveness of “consistent long-term policies” permanently reducing socioeconomic inequalities. It is curious to observe that, in the main text, the conclusion did not repeat this argument.

Thank you for this comment. We agree with the reviewer and we decided to exclude this sentence from the abstract.

---

## [Decision Letter · Decision Letter 1]

27 Oct 2022

Association of urban inequality and income segregation with COVID-19 mortality in Brazil

PONE-D-22-17397R1

Dear Dr. Susa Filho

We’re pleased to inform you that your manuscript has been judged scientifically suitable for publication and will be formally accepted for publication once it meets all outstanding technical requirements.

Kind regards,

Raphael Mendonça Guimaraes, PhD

Academic Editor

PLOS ONE

Additional Editor Comments (optional):

Reviewers' comments:

Reviewer's Responses to Questions

**Comments to the Author**

1. If the authors have adequately addressed your comments raised in a previous round of review and you feel that this manuscript is now acceptable for publication, you may indicate that here to bypass the “Comments to the Author” section, enter your conflict of interest statement in the “Confidential to Editor” section, and submit your "Accept" recommendation.

Reviewer #1: All comments have been addressed

2. Is the manuscript technically sound, and do the data support the conclusions?

Reviewer #1: Yes

3. Has the statistical analysis been performed appropriately and rigorously? 

Reviewer #1: Yes

4. Have the authors made all data underlying the findings in their manuscript fully available?

Reviewer #1: Yes

5. Is the manuscript presented in an intelligible fashion and written in standard English?

Reviewer #1: Yes

6. Review Comments to the Author

Reviewer #1: (No Response)

7. PLOS authors have the option to publish the peer review history of their article (what does this mean?). If published, this will include your full peer review and any attached files.

Reviewer #1: **Yes: **Jose Leopoldo Ferreira Antunes

---

## [Editor Report · Acceptance letter]

2 Nov 2022

PONE-D-22-17397R1 

Association of urban inequality and income segregation with COVID-19 mortality in Brazil 

Dear Dr. Sousa Filho:

I'm pleased to inform you that your manuscript has been deemed suitable for publication in PLOS ONE. Congratulations! Your manuscript is now with our production department. 

Kind regards, 

on behalf of

Dr. Raphael Mendonça Guimaraes 

Academic Editor

PLOS ONE